# Spatio-temporal modelling of lightning climatologies for complex terrain

Thorsten Simon[1,2], Nikolaus Umlauf[2], Achim Zeileis[2], Georg J. Mayr[1], Wolfgang Schulz[3], and Gerhard Diendorfer[3]

[1]Institute of Atmospheric and Cryospheric Sciences, University of Innsbruck, Austria
[2]Department of Statistics, University of Innsbruck, Austria
[3]OVE-ALDIS, Vienna, Austria

*Correspondence to:* T. Simon (thorsten.simon@uibk.ac.at)

**Abstract.** This study develops methods for estimating lightning climatologies on the $d^{-1}km^{-2}$ scale for regions with complex terrain and applies them to summertime observations (2010 – 2015) of the lightning location system ALDIS in the Austrian state of Carinthia in the Eastern Alps.

Generalized additive models (GAMs) are used to model both the probability of occurrence and the intensity of lightning. Additive effects are set up for altitude, day of the year (season) and geographical location (longitude/latitude). The performance of the models is verified by 6-fold cross-validation.

The altitude effect of the occurrence model suggests higher probabilities of lightning for locations on higher elevations. The seasonal effect peaks in mid July. The spatial effect models several local features, but there is a pronounced minimum in the Northwest and a clear maximum in the Eastern part of Carinthia. The estimated effects of the intensity model reveal similar features, though they are not equal. Main difference is that the spatial effect varies more strongly than the analogous effect of the occurrence model.

A major asset of the introduced method is that the resulting climatological information vary smoothly over space, time and altitude. Thus, the climatology is capable to serve as a useful tool in quantitative applications, i.e., risk assessment and weather prediction.

**Key words: lightning location data, generalized additive model, hurdle model, zero truncated Poisson distribution**

## 1 Introduction

Severe weather, associated with thunderstorms and lightning, causes fatalities, injuries and financial losses (Curran et al., 2000). Thus, the private and the insurance sector have a strong interest in reliable climatologies for such events, i.e., for risk assessment or as a benchmark forecast of a warning system.

Lightning is a transient, high-current (typically tens of kiloamperes) electric discharge in the air with a typical length of kilometers. The lightning discharge in its entirety is usually termed a *lightning flash* or just a *flash*. Each flash typically contains several *strokes* which are the basic elements of a lightning discharge (Rakov, 2016). Lightning flashes are rare events. Around 0.5–4 $km^{-2}yr^{-1}$ occur in the Austrian Alps (Schulz et al., 2005). Lightning location data homogeneously detected—with the

same network and selection algorithm—typically cover on the order of 10 years. Consequently not enough data are available to compute a spatially resolved climatology on the $km^{-2}$ scale by simply taking the mean of each cell—even less so if a finer temporal resolution then $yr^{-1}$ is desired, which is the case for lighting following a prominent annual cycle. Thus there is the need to develop methods to robustly estimate lightning climatologies by exploiting information contained not only in each

5 analysis cell and maybe its neighboring cells but in the complete data set. Lightning might, e.g., depend on the altitude of the grid cells so that combining the information from all cells at a particular altitude band increases the sample size and leads to a more robust estimate. Other common effects that might be exploited are geographical location, day of the year, time of day, slope orientation, distance from the nearest mountain ridge, . . .

One possibility of harnessing the complete data set to produce a lightning climatology in such a manner are generalized

additive models (GAMs, see, Hastie and Tibshirani, 1990; Wood, 2006). They can include these common effects as additive terms. Each of these term might be of arbitrary complexity and represent the potentially nonlinear relationship between lightning and the covariate. An additional advantage of GAMs is the ability for inference. It can be tested whether each of the included effects *is* actually an effect—or not significant and thus not exploiting common information in the whole data set. GAMs allow the inclusion of expert knowledge through the choice of the covariates. GAMs have been used to compile clima-

tologies of extremes (Chavez-Demoulin and Davison, 2005; Yang et al., 2016) and full precipitation distributions (Rust et al., 2013; Stauffer et al., 2016). A further benefit of GAMs is that all the parameters of a distribution, e.g., scale and shape, can be modeled not only the expected value (Stauffer et al., 2016).

In this study GAMs are applied to estimate a climatology of the probability of lightning and a climatology of the expected numbers of flashes with a spatial resolution of 1 $km^2$ and a temporal resolution of 1 day. They serve also as proxies for clima-

20 tologies of the occurrence of thunderstorms (e.g., Gladich et al., 2011; Poelman, 2014; Mona et al., 2016) and thunderstorm intensity, respectively. The climatologies will be compiled for a region in the southeastern Alps.

A study investigating lightning data for the period 1992 to 2001 (Schulz et al., 2005) found that flash densities over the complex topography of Austria vary between 0.5 and 4 flashes $km^{-2}yr^{-1}$ depending on the terrain. Data from the same lightning location system (LLS) was also analyzed to obtain thunderstorm tracks (Bertram and Mayr, 2004). The key finding

is that thunderstorms are often initialized at mountains of moderate altitude and propagate towards flat areas afterwards.

Other studies focus on lightning detected in the vicinity of the Alps: A 6-year analysis of lightning detection data over Germany reveals highest activity in the northern foothills of the Alps and during the summer months, where the number of thunderstorm days goes up to 7.5 $yr^{-1}$ (Wapler, 2013). Lightning activity is also high along the southern rim of the Alps. Feudale et al. (2013) found ground flash densities up to 11 flashes $km^{-2}yr^{-1}$ in northeast Italy, which is south of our region

of interest, Carinthia.

The manuscript is structured as follows: The lightning detection data, the region of interest, Carinthia, and the pre-processing of the data are described in Sect. 2. The methods to estimate the lightning climatologies from the data are based on generalized additive models (Sect. 3). The nonlinear effects estimated for occurrence and intensity model and exemplary climatologies are presented afterwards (Sect. 4). Some special aspects of interest for end-users are discussed in Sect. 5. The study is summarized

and concluded at the end of the manuscript (Sect. 6).

## 2 Data

In this study 6 years of data (2010 – 2015) from the ALDIS detection network (Schulz et al., 2005) are included. The data within this period are processed in real-time by the same lightning location algorithm ensuring stationarity with respect to data processing. The summer months May to August are selected, as this is the dominant lightning season in the Eastern Alps. The original ALDIS data contains single strokes and information which strokes belong to a flash. In order to analyze flashes, solely the very first stroke of a flash is taken into account. Both cloud-to-ground and intra-cloud lightning strikes are considered, as both typically indicate thunderstorms which are of interest in this study.

ALDIS is part of the European cooperation for lightning detection (EUCLID) (Pohjola and Mäkelä, 2013; Schulz et al., 2016), which bundles European efforts in lightning detection. Schulz et al. (2016) present an evaluation of the performance of lightning detection in Europe and the Alps by comparison against direct tower observations with respect to detection efficiency, peak current estimation and location accuracy. The median location error was found to be in the order of $100\ m$. Furthermore, they show that the flash detection efficiency is greater than 96% (100%) if one of the return strokes in a flash had a peak current greater than 2kA (10kA). However, it is impossible to determine the detection efficiency of intra-cloud flashes without a locally installed VHF network. Thus no attempt made in Schulz et al. (2016) to characterize the detection efficiency of intra-cloud flashes.

The region of interest is the state of Carinthia in the south of Austria at the border to Italy and Slovenia. Carinthia extends $180\ km$ in west-east direction and $80\ km$ in south-north direction. The elevation varies between $339\ m$ to $3798\ m$ above mean sea level (a.m.s.l.). For invoking elevation as a covariate into the statistical model (Sect. 3) digital elevation model (DEM) data (Kärnten, 2015), which is on hand on a $10\ m \times 10\ m$ resolution, is averaged over $1\ km \times 1\ km$ cells (Fig. 1), which leads to a maximum elevation of $3419\ m$ a.m.s.l. with respect to this resolution. As the distribution of the altitude is highly skewed, the logarithm of the altitude serves as covariate, which is distributed more uniformly in the range from 6 to 8.

The lightning data, for May to August of the 6 years, are transferred to the same $1\ km \times 1\ km$ raster by counting the flashes within one spatial cell and per day. This procedure yields 9904 cells and 738 days for a total of $n = 7309152$ data points, from which 157440 (2.15%) show lightning activity. The amount of cells in which a specific number of flashes was detected decreases rapidly for increasing count numbers. The most extreme data point has 37 flashes per cell and day (Fig. 2). The mean number of detected flashes in the cells *given* lightning activity is 1.75.

Figure 3 shows an example for a climatology based on empirical estimates for July. Here the number of days with lightning is divided by the total number of days for every single grid cell. While some patterns emerge, a large amount of noise is visibly superimposed.

## 3 Methods

This section introduces the statistical models for estimating the climatologies for lightning occurrence and lightning intensity. The aim of the statistical model is to explain the response, i.e., the probability of occurrence or counts of flashes, by appropriate spatio-temporal covariates, i.e., logarithm of the altitude (`logalt`), day of the year (`doy`) and geographical location (`lon`, `lat`).

Since the response might nonlinearly depend on the covariates we choose generalized additive models (GAMs) as a statistical framework, for which a brief introduction is presented in Sect. 3.1.

It is assumed that the number of flashes detected within a cell and day are generated by a random process $Y$. Realizations of the random process are denoted by $y_i \in \{0, 1, 2, \ldots\}$, where $i = 1, 2, \ldots, n$ indicates the observation. Two distinct models are set up: first, a model for the probability of the occurrence of lightning $\Pr(Y > 0)$ within a cell and a day; second, a truncated count model to assess the expected number of flashes within a cell *given* lightning activity $E[Y|Y > 0]$. This procedure refers to a hurdle model (Mullahy, 1986; Zeileis et al., 2008) which has the further benefit to be able to handle the large amount of zero valued data points (97.85%, in our case). The occurrence model and the intensity model are specified in Sect. 3.2 and Sect. 3.3, respectively.

In Sect. 3.4 a short overview over the applied verification techniques is given, i.e., cross-validation, scoring rules and boot-strapping.

## 3.1 Generalized additive model

The main motivation for using a GAM is the possibility to estimate (potentially) nonlinear relationships between the response and the covariates. In the following, the basic concept of GAMs is introduced for an arbitrary parameter $\theta$ of some probability density function $d(\cdot; \theta)$ (PDF). A GAM aiming at modelling a spatio-temporal climatology over complex terrain would have the form,

$$g(\theta) = \beta_0 + f_1(\texttt{logalt}) + f_2(\texttt{doy}) + f_3(\texttt{lon}, \texttt{lat}), \tag{1}$$

where $g(\cdot)$ is a link function that maps the scale of the parameter $\theta$ to the real line. The right hand side is called the additive predictor, where $\beta_0$ is the intercept term and $f_j$ are unspecified (potentially) nonlinear smooth functions that are modeled using regression splines (Wood, 2006; Fahrmeir et al., 2013). For each $f_j$ a design matrix $X_j$ containing spline basis functions is constructed. Thus the GAM can be written as generalized linear model (GLM),

$$g(\theta) = \beta_0 + \sum_{j=1}^{3} X_j \beta_j. \tag{2}$$

The coefficients $\beta = (\beta_0, \beta_1^\top, \beta_2^\top, \beta_3^\top)$ are estimated by maximizing the penalized log-likelihood,

$$l(\beta) = \sum_{i=1}^{n} \log(d(y_i; g^{-1}(\beta_0 + \sum_{j=1}^{3} X_j \beta_j))) - \frac{1}{2} \sum_{j=1}^{3} \lambda_j \beta_j^\top S_j \beta_j, \tag{3}$$

where the first term on the right-hand side is the unpenalized log-likelihood. The second term is added to prevent overfitting by penalizing too abrupt jumps of the functional forms. $\lambda_j$ are the smoothing parameters corresponding to the functions $f_j$, respectively. For $\lambda_j = 0$ the log-likelihood is unpenalized with respect to $f_j$. When $\lambda_j \to \infty$ the fitting procedure will select a linear effect for $f_j$. The selection of the smoothing parameters $\lambda_j$ is performed by cross-validation (Sect. 3.4).

The value of $\lambda_j$ determines the degrees of freedom of the associated effect. Lower values of $\lambda_j$, e.g., small penalization, lead to an effect with more degrees of freedom, which might explain more features but is also prone to overfitting. High values of $\lambda_j$,

e.g., strong penalization, result in an effect with fewer degrees of freedom. Thus fewer features can be explained. The balance between small and strong penalization and thus the corresponding degrees of freedom is found by performing cross-validation (Sect. 3.4).

$S_j$ in Eq. 3 are prespecified penalty matrices, which depend on the choice of spline basis for the single terms. The reader is referred to Wood (2006) for more details.

Estimation of a GAM for such a large dataset, i.e., 7309152 data points, is feasible, e.g., via function `bam()` (for *big additive models*) implemented in the **mgcv** package (Wood et al., 2015; Wood, 2016) of the statistical software R (R Core Team, 2016).

## 3.2 Occurrence model

The first component models the probability of lightning $\Pr(Y > 0) = \pi$ to occur within a $1\ km \times 1\ km$ cell and a day. The Bernoulli distribution with the parameter $\pi$ of the PDF,

$$d_{\mathrm{Be}}(y; \pi) = \pi^y (1 - \pi)^{1-y},\tag{4}$$

will be fitted. Since the data is binomial $y \in \{0, 1\}$ indicates no lightning and lightning within the cells, respectively. The model for $\pi$ takes the form of Eq. 1 with $\pi$ replacing $\theta$. The complementary log-log $g(\pi) = \log(-\log(1 - \pi))$ is implemented as link function.

## 3.3 Intensity model

The second part is the truncated count component for the expected number of flashes *given* lightning activity. We will refer to this component as the intensity model. It is assumed that the positive counts of flashes within a spatial cell and day follow a zero truncated Poisson distribution with the PDF,

$$d_{\mathrm{ZTP}}(y; \mu) = \frac{d_{\mathrm{Pois}}(y, \mu)}{1 - d_{\mathrm{Pois}}(0, \mu)}\tag{5}$$

where $y \in \{1, 2, \dots\}$, and $d_{\mathrm{Pois}}(\cdot; \mu)$ is the PDF of the Poisson distribution with expectation $\mu$. The conditional expectation is $\mathrm{E}[Y|Y > 0] = \mu/(1 - e^{-\mu})$. The GAM for this component has the form of Eq. 1 with $\mu$ replacing $\theta$. The logarithm serves as link function $g(\mu) = \log(\mu)$.

The family for modelling the zero truncated Poisson distribution `ztpoisson()` within a GLM or GAM framework is implemented in the R-package **countreg** (Zeileis and Kleiber, 2016). For more information on and a formal definition of hurdle models the reader is referred to Zeileis et al. (2008).

## 3.4 Verification

In this section the verification procedures are briefly introduced, namely the cross-validation, the applied scores and the block-bootstrapping.

In order to ensure the verification of the model along independent data, we applied a 6-fold cross-validation (Hastie et al., 2009). 6 years of data are available. The parameters of the model are estimated based on 5 years of the data and validated on the remaining year. This is done 6 times with every single year serving as validation period once.

The log-likelihood is applied as scoring function, which is also called logarithmic score in the literature on proper scoring rules (Gneiting and Raftery, 2007).

To assess confidence intervals of the estimated parameters and effects, *day-wise block-bootstrapping* was performed. With *day-wise block-bootstrapping* we mean the following: We resample the 738 dates of all available days with repetition and pick all the data observed on these days spatially. This procedure is executed 1000 times in order to assess confidence intervals.

## 4   Results

This section presenting the results of the statistical models is structured as follows: first, the nonlinear effects of the occurrence model are described in Sect. 4.1; second, the effects of the intensity model are presented in Sect. 4.2. Finally, exemplary applications illustrate in Sect. 4.3 how climatological information can be drawn from the models.

### 4.1   Occurrence model

The estimates of the effect of the occurrence model (Sect. 3.2) are depicted in Fig. 4. The values are on the scale of the additive predictor, i.e. the right hand side of Eq. 1. The additive predictor, takes the value of the intercept term $\beta_0$ if the sum of all other effects is equal to zero. Its estimate is $\beta_0 = -3.97 \, (-4.15, -3.80)$ on the complementary log-log scale, which is $1.87\%$ $(1.56\%, 2.21\%)$ in terms of probability of lightning. The numbers in parentheses are the 95% confidence intervals computed from 1000 day-wise block-bootstrapping estimates.

How the effects in Fig. 4 can be interpreted to obtain the probability of lightning at a particular location and day is shown for the location E (Rosennock in Fig. 1) and July 20. The location is at an altitude of $2440 \, m$ (Table 1), for which the altitude effect is roughly $0.34$ (Fig. 4a). The contribution of the seasonal effect (Fig. 4b) for July 20 is about $0.64$. The spatial effect (Fig. 4c) has a value of $-0.03$ at this geographical location ($13.71° \, \mathsf{E}, 46.88° \, \mathsf{N}$). Adding these values to the intercept $\beta_0 = -3.97$ yields $-3.02$, which is on the scale of the additive predictor. It needs to be transferred with the inverse of the complementary log-log function to obtain the value in probability space, which is $4.76\%$.

The second term $f_1(\mathtt{logalt})$ of the additive predictor models the effect of the logarithm of the altitude. $f_1(\mathtt{logalt})$ varies from roughly $-0.2$ for low altitudes to values greater than $0.5$ for altitudes above $2800 \, m$ (Fig. 4a). This function takes $7.9$ degrees of freedom. Its shape is close to exponential, which suggests that a linear term for the altitude might be sufficient. For altitudes above $2000 \, m$, however, the nonlinear term leads to larger values than a linear term $\beta_1 alt$ would do.

The temporal or seasonal effect $f_2(\mathtt{doy})$, i.e., the dependence of the target on the day of the year ($\mathtt{doy}$), shows a steep increase during May, reaches its maximum in mid July and decreases slowly during August (Fig. 4b). This result indicates that the main lightning season in Carinthia lasts from mid June until end of August. The estimated degrees of freedom are $2.5$, which leads to the simple shape of the seasonal effect with one clear maximum.

The spatial effect $f_3(\texttt{lon},\texttt{lat})$, which explains the spatial variations of the linear predictor that cannot be explained by the altitude term $f_1(\texttt{logalt})$, requires 138 degrees of freedom. (Fig. 4c). Most prominent features are the minimum near the northwest and the maximum in the mid to eastern part of Carinthia. In the northwest the highest mountains, the High Tauern, of Carinthia are located. The minimum with values less than $-0.3$ on the complementary log-log scale suggests that lightning activity is less pronounced in this region. This finding is in line with former analyses of the lightning activity in Austria (Troger, 1998; Schulz et al., 2005), which stated that the main alpine crest is an area with a minimum in flash density. The maximum zone with values exceeding $0.3$ in the mid to eastern part of Carinthia covers the so-called Gurktal Alps. In comparison with the High Tauern, the Gurktal Alps have a lower average elevation and the mountains are not as steep. Such maxima at moderate or low altitude are mostly modeled by the spatial effect, not by the altitude effect.

As the altitude is a function of longitude and latitude, one could ask whether it would be sufficient take only a spatial effect into account that implicitly contains the altitude and skip the explicit altitude effect. In general the presented method would be capable to model the influence of the altitude within the spatial effect implicitly. However, the shape of the altitude in the region of interest is very complex. Thus, a spatial effect with a large degree of freedom would be required in order to account for the complex altitude shape. As we know the shape of the altitude we can pass it to the GAM as an isolated effect. The altitude effect contains only information associated with the altitude while the remaining effects are captured by the spatial term.

The introduced model (Eq. 1) could also be extended by potentially nonlinear functions of other covariates meaningful for a climatological assessment, e.g., surface roughness, slope and aspect of topography. However in the present case adding these covariates was not improving the model.

## 4.2  Intensity model

The nonlinear effects of the intensity model (Sect. 3.3) are depicted in Fig. 5. The estimate of the intercept term takes the value $\beta_0 = -0.01 \,(-0.19, 0.14)$ which leads to a expected number of flashes *given* lightning activity of $1.57 \,(1.47, 1.68)$ when the sum of all other effects is equal to zero.

The altitude effect $f_1(\texttt{logalt})$ (Fig. 5a), with $5.4$ degrees of freedom, reveals a similar functional form as the altitude effect of the occurrence model (Fig. 4a). However, it has a flatter shape for the terrain between $600\,m$–$1200\,m$ and a steeper increase for high altitudes above $2000\,m\;a.m.s.l.$.

The seasonal effect $f_2(\texttt{doy})$ is $-0.5$ in early May, reaches a maximum of $0.3$ in early July and decreases to values around $-0.3$ until the end of August (Fig. 5b). Thus the amplitude of this effect is not as strong as the seasonal effect of the occurrence model (Fig. 4b) and the location of the maximum is earlier. It has $2.1$ degrees of freedom.

The spatial effect $f_3(\texttt{lon},\texttt{lat})$ varies strongly and requires 166 degrees of freedom which is more than the corresponding effect of the occurrence model. However, there are some features common for both effects. For instance, the prominent maximum visible in Fig. 5c in the Gurktal Alps appears also in the spatial effect of the count model (Fig. 4c). Common is also the strong minimum in the western part of the domain. The most pronounced new feature is the strong local maximum with values exceeding $0.9$ in the south of Carinthia. A $165\,m$ radio tower is installed on the peak of the Dobratsch mountain (location

C in Fig. 1), which triggers lightning strokes under suitable conditions, i.e., occurrence of a thunderstorm. Other maxima of this effect could also be attributed to sites of radio towers, which suggests that the number of flashes is more sensitive to local constructions than the probability of lightning.

## 4.3 Applications

In order to illustrate how climatological information can be drawn from the GAMs, two different kinds of applications are presented. First, maps show spatial climatologies (Fig. 6 and Fig. 7). Here, the occurrence model and/or the intensity model are evaluated for one specific day. Second, the seasonal climatology for selected $1\,km \times 1\,km$ grid cells are discussed (Fig. 8). Here, the models are evaluated with respect to the geographical location of the point of interest and its altitude.

The spatial distribution of climatological probabilities of lightning to occur in a cell for July 20 (close to the seasonal peak)
varies from $1.8\%$ to $6.5\%$ (Fig. 6). In the western part of the domain, local valleys and mountain ridges become visible through the altitude effect (Fig. 4a). However, the highest probabilities do not occur over the highest terrain in the northwest, where the spatial effect counteracts the altitude effect leading to moderate probabilities around $2\%$ to $3\%$. The spatial effect (Fig. 4c) is responsible for the maximum over the moderate altitude region of the Gurktal Alps. Such a map can also serve as thunderstorm climatology when lightning is taken as a proxy for thunderstorms.

A comparison of the Figures 6 and 3 illustrates some of the benefits of using GAMs instead of taking averages in each grid cell for computing expected values of lightning occurrence. Harnessing the information from the complete data set instead of using only information contained in each grid cell removes the noise and makes the overall pattern visible, e.g., the difference between lightning over valleys and ridges.

For the same day, July 20, the expected number of flashes is depicted in Fig. 7. This is the product of probabilities of lightning
$\pi$ from the occurrence model and the expected number of flashes *given* lightning activity, which is derived from the intensity model. Values are ranging from $0.028$ to $0.166$. The lowest values can be found in the northwestern part of Carinthia where the spatial effects of both models reveal a minimum. Next to the maximum in the Gurktal Alps, where also maxima in the spatial effects of both models can be found, a second peak appears at the Dobratsch mountain (location C in Fig. 1) which is due to the local maximum in the spatial effect of the intensity model (Fig. 5c).

Next to the spatial information one can extract seasonal climatologies for different locations (Fig. 8). These are computed exemplary for five sites (Table 1). Fig. 8a shows the climatologies of lightning probability. Differences between the annual cycles of the probabilities are due to the altitude effect and the spatial effect of the occurrence model (Fig. 4). The highest probabilities between $4\%$ and $5\%$ are modeled in July for location B (dashed line), which is located at the southwestern border of Carinthia in vicinity of a local maximum of the spatial effect (Fig. 4c). This climatology exhibits a strong seasonality, as
probabilities fall below the $1\%$ level. Though located at a similar altitude, the climatology of location A (solid line) reveals maximum values less than $2\%$. This difference is due to the spatial effect, which exhibits a clear minimum in northwestern Carinthia. The climatology of location D (dashed dotted line) in the lower plains in the eastern part of Carinthia show moderate chance of lightning with values around $3\%$ during the peak of the season.

The climatologies of the expected number of flashes are depicted in Fig. 8b. The order of location has changed. In particular the highest number of flashed are expected for location C which is the Dobratsch mountain. This is caused by the strong local maximum in the spatial effect of the intensity model (Fig. 5c). The legend shows expected number of flashes accumulated over the lightning season, which leads to values between 2.1 for location A and 7.6 for location C. These values are in good agreement with the analysis by Schulz et al. (2005, Fig 5. therein).

Finally, it is also possible to derive relative frequencies of the number of flashes of a specific location and day of the year from the GAM. The relative frequencies have been derived for the 5 sample locations (Table 2). The first column of the table with the probabilities for no flashes to occur contains only information from the occurrence model (cf. Fig. 8a). All other probabilities for one or more flashes are derived from both the occurrence and the intensity model. The influence of the intensity model is especially dominant in the relative frequencies for location C, where the probability of having 4 or more flashes on July 20 is 1.54%.

## 5  Discussion

This section addresses two points helpful for end-users. The first one is on how to choose the cross-validation score in order to avoid overfitting of the seasonal effect (speaking technically the selection of its smoothing parameter $\lambda$). The second point is a discussion on how the introduced model (Eq. 1) can be extended towards a weather prediction tool, i.e., for warning purposes.

For illustration of the first point a subset of the large dataset is selected. We pick all data points in a $5 \times 5$ neighborhood around the location E. Thus, only $6 \; years \times 123 \; days \times 25 \; cells = 18450 \; data \; points$ remain. The probability of lightning $\pi$ is the target variable. Furthermore, altitude and spatial effects are omitted for simplicity for such a small region (cf. the smooth spatial effect of the occurrence model in Fig. 4c). Thus the GAM has the form,

$$g(\pi) = \beta_0 + f(\texttt{doy}). \tag{6}$$

The model is fitted twice: first, with the selection of the smoothing parameter $\lambda$ by six-fold cross-validation where the observations made on a single day are kept together in a block, e.g., the cross-validation splits the $6 \; years \times 123 \; days = 738 \; days$ into six parts; second, $\lambda$ is determined by six-fold cross-validation without the *day-wise* blocks, e.g., the cross-validation splits the $18450 \; data \; points$ randomly into six parts. In both cases the maximal number of degrees of freedom is set to 30.

Fig. 9 shows the estimates of the two models. The estimate resulting from the cross-validation with *day-wise* blocks is much smoother (1.9 degrees of freedom) than the estimate resulting from the cross-validation without daily blocks (29.9 degrees of freedom). Thus the latter estimate takes roughly the maximal degree of freedom and is obviously overfitted.

The reason for the distinct estimates lies in the dependence structure of the data. For one cell the probability to detect lightning on one day *given* lightning was detected on the previous day is 6.7%. Spatial dependence is much stronger. Provided that lightning occurs in one cell, the probability of lightning to occur in the adjacent cell is 41%. This strong spatial dependence comes with a physical meaning. First, the preconditions for thunderstorms and lightning to take place vary much stronger from day-to-day than in the course of a single day. Second, thunderstorm systems, i.e., multi-cell thunderstorms or super-cell thunderstorms, cover a large area or even travel over a larger area (Markowski and Richardson, 2011).

For this reason we recommend to explore the dependence structure of the data first and to define the cross-validation score according to this dependence structure.

Finally, we discuss how the introduced model (Eq. 1) can be extended in order to serve as a weather prediction tool. It is possible to add further predictors from a numerical weather prediction system to the right hand side of Eq. 1. In the case of lightning and thunderstorms suitable predictors could be convective inhibition energy (CIN), convective available potential energy (CAPE), vertical shear of horizontal winds or large scale circulation patterns (e.g., Bertram and Mayr, 2004; Chaudhuri and Middey, 2012). Within the GAM framework nonlinear effects and interactions of these predictors can be modeled. Another major benefit of this procedure is that the climatology is nested within the additive predictor. Thus the performance of the prediction tool would be at least on the quality level of the climatology, but would not fall below.

## 6 Conclusions

This study presented how generalized linear models (GAMs) (e.g., Hastie and Tibshirani, 1990; Wood, 2006) provide a useful tool for building a lightning climatology or a climatology for the occurrence of thunderstorms. The main concept is to decompose the signal into different effects: an altitude effect, a seasonal effect and a spatial effect. The most beneficial aspect of this method is that smooth estimates for these effects are obtained on such a fine spatial and temporal scale as $1~km^2$ and 1 day. This makes the resulting climatology a valuable tool for quantitative purposes, e.g., risk assessment or benchmarking in weather prediction. In order to provide smooth effects the method harnesses information from the complete data set not just separately in each cell as would be the case by simply averaging the data. Even more effects than demonstrated in this paper can be included, e.g., slope and aspect of topography or parameters associated with land use. The choice of common effects allows to include expert knowledge. Additionally and importantly, applying GAMs will also show which of these proposed effects is significant.

A hurdle approach was employed to compute a climatology of the *intensity* of lightning in order to properly handle the large amount of zeros in the data. Thus two aspects of lightning are captured by the models: the probability of lightning to occur and the number of flashes detected within a grid cell. The effects of the two models are similar though not equal. In particular the spatial effect of the intensity model varies more strongly than the corresponding effect of the occurrence model. For instance, local intensity maxima are triggered in vicinity of radio towers.

In sum, the occurrence model and the count model took roughly 150 and 180 degrees of freedom, respectively. This is a relatively small number compared to the degrees of freedom required by other methods. Counting and averaging flashes with respect to a resolution of $km^{-2}yr^{-1}$ would lead to 9904 degrees of freedom in the introduced case without capturing the seasonal cycle. Thus the GAM approach leads to a smooth, nonlinear and sparse quantification of the climatologies.

*Author contributions.* Thorsten Simon, Georg J. Mayr and Achim Zeileis defined the scientific scope of this study. Thorsten Simon performed the statistical modelling, evaluation of the results and wrote the paper. Georg J. Mayr supported on the meteorological analysis. Nikolaus Umlauf and Achim Zeileis contributed to the development of the statistical methods. Wolfgang Schulz and Gerhard Diendorfer

were in charge of data quality and advised on the lightning related references. All authors discussed the results and commented on the manuscript.

*Acknowledgements.* We acknowledge the funding of this work by the Austrian Research Promotion Agency (FFG) project *LightningPredict* (Grant No. 846620). The computational results presented have been achieved using the HPC infrastructure LEO of the University of Innsbruck. Furthermore we are deeply grateful to the editor and reviewers for their valuable comments.

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

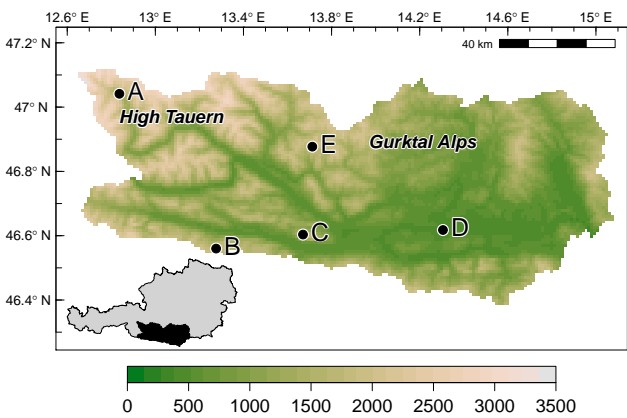

**Figure 1.** Altitude of Carinthia ($m\ a.m.s.l.$) averaged over $1\ km \times 1\ km$ cells. Attributes of sample locations are listed in Table 1. The location map shows the location of Carinthia (black) in Austria (lightgray).

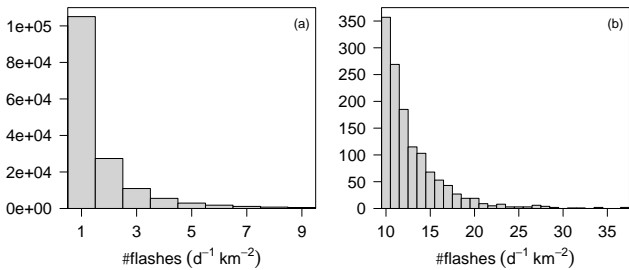

**Figure 2.** Daily frequency of $1\ km \times 1\ km$ grid cells with counts of flashes (excluding zeros). The right panel (b) shows a zoom into the tail of the distribution. The percentage of boxes with no flashes detected is $97.85\%$.

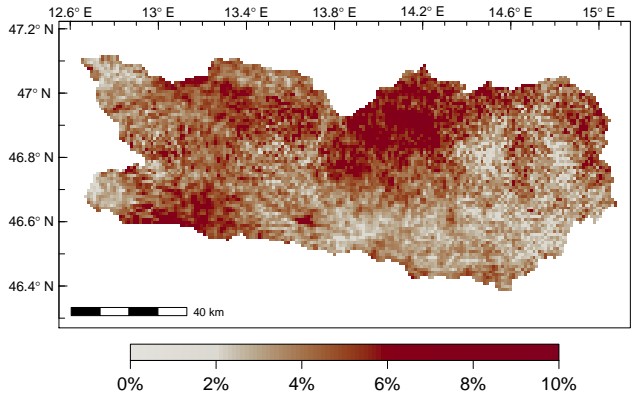

**Figure 3.** Empirical climatological probability of lightning for a day in July in Carinthia on the $1\ km \times 1\ km$ scale computed from counting the days with lightning over all July days in the six year period and dividing by the number of all July days.

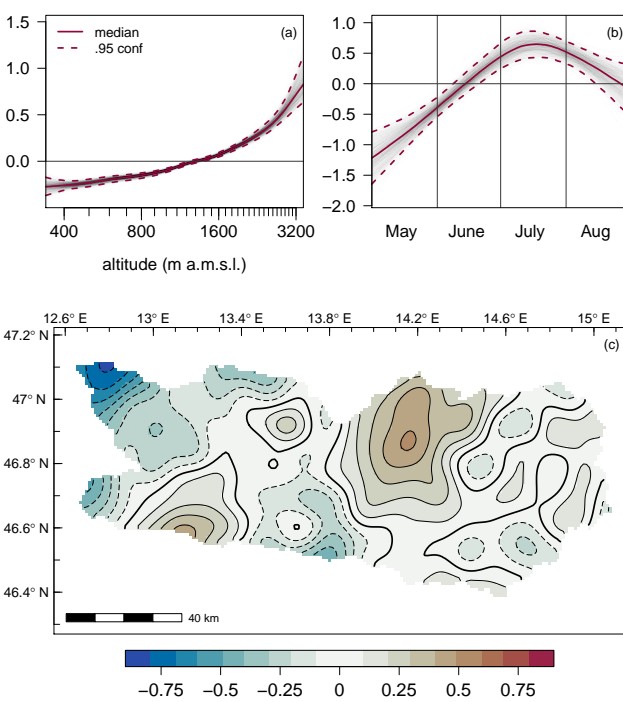

**Figure 4.** The effects of the occurrence model on the scale of the additive predictor. **a:** The altitude (`logalt`) effect. Ticks on the x-axis are set in 100 m intervals. The gray lines show 1000 estimates from day-wise block-bootstrapping. The solid red line is the median of the 1000 estimates, the dashed red lines are the 95% confidence intervals. **b:** The seasonal (`doy`) effect. **c:** The spatial (`lon`,`lat`) effect. The plot shows the median of 1000 estimates from day-wise block-bootstrapping. The difference between two contour lines is 0.1. Dashed contour lines indicate negative values.

| id | name | lon ($^\circ$ E) | lat ($^\circ$ N) | alt ($m\ a.m.s.l.$) |
|----|------|---------|---------|--------------|
| A | Heiligenblut | 12.84 | 47.04 | 1315 |
| B | Nassfeld | 13.28 | 46.56 | 1525 |
| C | Dobratsch | 13.67 | 46.60 | 2166 |
| D | Klagenfurt | 14.31 | 46.62 | 447 |
| E | Rosennock | 13.71 | 46.88 | 2440 |

**Table 1.** Coordinates of the sample locations in Figure 1.

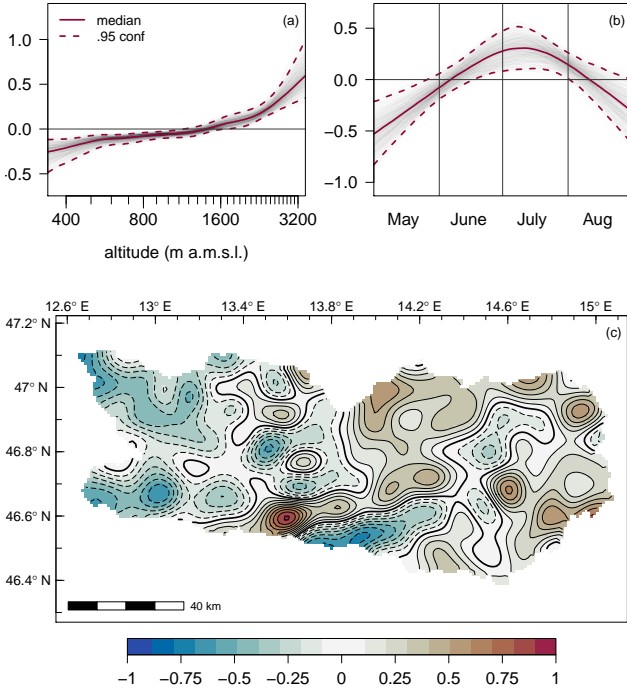

**Figure 5.** The effects of the intensity model on the scale of the additive predictor. Labeling is analog to Fig. 4. **a:** The altitude (`logalt`) effect. **b:** The seasonal (`doy`) effect. **c:** The spatial (`lon, lat`) effect.

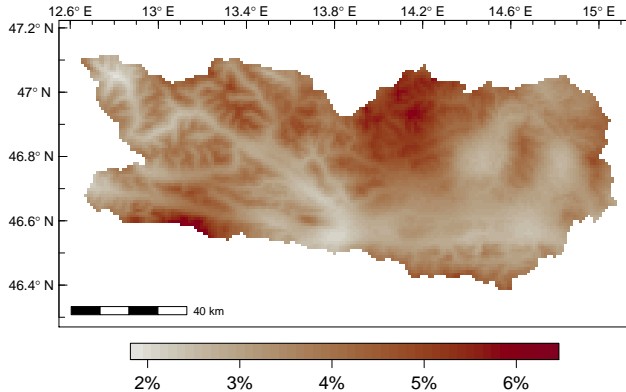

**Figure 6.** Climatological probability (expected values) of lightning in Carinthia on the $1~km \times 1~km$ scale for July 20 computed with a generalized additive model (GAM).

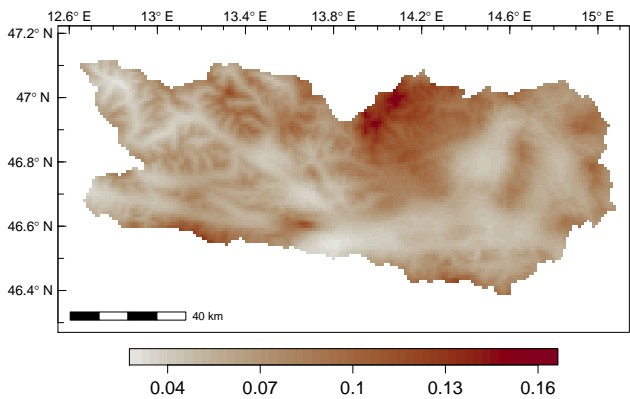

**Figure 7.** Climatological number of flashes (expected values) in Carinthia on the $1\ km \times 1\ km$ scale for July 20.

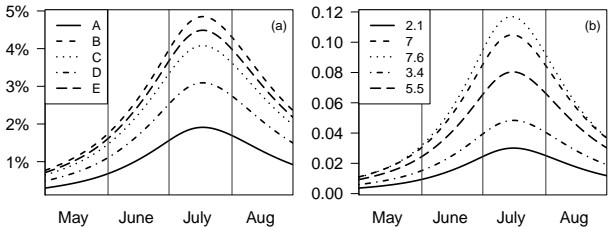

**Figure 8.** Seasonal climatologies for sample locations, which are highlighted in Figure 1. **a:** Occurrence model. **b:** Expected number of flashes. The legend shows expected number of flashes accumulated over the lightning season.

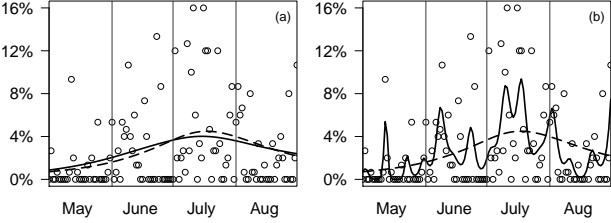

**Figure 9.** Local fits for the location E. Circles show *empirical* estimates. For comparison the estimate of the full occurrence model is added (dashed line). **a:** Solid line is the GAM evaluated by cross-validation with *day-wise* blocks. **b:** Solid line is the GAM evaluated by cross-validation without *day-wise* blocks.

|   | 0     | 1    | 2    | 3    | >4   |
|---|-------|------|------|------|------|
| A | 98.16 | 1.12 | 0.52 | 0.16 | 0.04 |
| B | 95.37 | 1.74 | 1.49 | 0.86 | 0.54 |
| C | 95.54 | 0.76 | 1.10 | 1.06 | 1.54 |
| D | 96.94 | 1.81 | 0.88 | 0.28 | 0.09 |
| E | 95.24 | 2.24 | 1.52 | 0.69 | 0.31 |

**Table 2.** Relative frequencies (%) of number of flashes (columns) for July 20 of the sample locations (rows) in Figure 1 derived from the occurrence and intensity GAM.