# Peer review of "Spatio-temporal modelling of lightning climatologies for complex terrain"

_Natural Hazards and Earth System Sciences, 2016_

## Referee Comment (RC1) · Anonymous Referee #1 · 1 Aug 2016

I have read this paper dealing with the lightning climatology in Austria. While the paper is well written, clear and at a high level of English, I am not sure why a model is needed to describe the lightning climatologies, when the raw data is already available to the authors. The authors state that for risk assessment or when climatology is used as a benchmark weather forecast this model will be valuable. But why do we need a model when we have the actual real lightning climatology. If we need to know what the probability is of lightning hitting location A, we can calculate this from the raw data.

In addition, the model is developed using the lightning data itself, and then tries to predict the same lightning data. So the model input and output are not independent of each other. A correct model should use parameters A, B and C to predict D. not A, B and D to predict D. Furthermore, the model should be developed for a specific period, i.e. 1992 to 2000 (for example) and then tested on the year 2001 to see if the model

can reproduce the lightning of 2001. In fact, it would be interesting and valuable to compare the model output (2001) with the real data (2001). How well correlated are the lightning estimates by the model for 2001 (based on a model constructed with input data from 1992-2000) with the real lightning from 2001. That is a legitimate test of the model.

Finally, if the model is a physical model, then it should be applicable to other regions of Austria. How well does this model predict lightning in other regions of Austria (or Europe)? If it is only good for Carinthia, then why bother? Just use the real observed data for risk maps.

Specific comments: Page 2, Line 2: of Austria vary

Line 24: data are

line 30: period are

Page 3, line 1: What is the detection efficiency of the intra-cloud flashes relative to the CG flashes?

line 6: what about the detection efficiency in %?

line 13: data are

line 24: Is it a correct assumption to assume a random process?

Page 4: This reminds me of the KISS principle.....(Keep it Simple, Stupid)

Page 5, line 27: The raw data also shows that the main lightning season is from June until end of August. So what is so great about this model? Why do we need it? To tell us something we already know from the raw data? I don't understand the logic behind the model. What can it tell us that we don't already know.

Page 8, line 21: But if we HAVE the climatology, why do we need this tool?

line 24: I do not understand why this is a useful too when the raw data give a better

estimate of the climatology.

line 26: smooth estimates can also be obtained by averaging the raw data temporally and spatially.

line 27: Why not simply use the real raw data? I do not understand why a model is needed.

Page 12, Figure 2 caption: cells with

Page 13, Figure 3: What are units of y-axis in upper plots?

Page 14, Figure 5: How does this differ from the raw data climatology. Maybe show one next to the other.

Page 15, Figure 7: What are the numbers in the key of the figure on right. 2.1? 7?....

---

## Author Comment (AC1) · 8 Aug 2016

Thank you very much for your informative and detailed comments. It's obvious that we will have to work on two aspects of the paper during the revision process: firstly, We have to motivate the need for this method better and secondly we have to present the method in a more accessible way.

As for the motivation:

The main motivation to process the raw data by a statistical model is to improve the signal-to-noise ratio. The raw lightning data contains a lot of noise due to the high variability of processes generating lightning. In general this is not only true for lightning, but also for most other atmospheric variables, e.g. precipitation, wind speed and direction, etc. The GAM applied in our study aims at filtering effects/signals associated

with altitude, day of the year and space and to separate these from the noise. We will extend the introduction in the next version of the manuscript with respect to this point in order to enhance the motivation of the method.

Furthermore, we have to present the method in a more accessible way for readers unfamiliar with advanced regression methods. We are aware that the mathematics behind the methods is complex and maybe even daunting for readers with no or little statistical background. However, we are willing to work on that issue, i.e., presenting the method in a more accessible way, in order to encourage more scientists from the lightning community to work with GAMs.

–

General point 1: I have read this paper dealing with the lightning climatology in Austria. While the paper is well written, clear and at a high level of English, I am not sure why a model is needed to describe the lightning climatologies, when the raw data is already available to the authors. The authors state that for risk assessment or when climatology is used as a benchmark weather forecast this model will be valuable. But why do we need a model when we have the actual real lightning climatology. If we need to know what the probability is of lightning hitting location A, we can calculate this from the raw data.

Answer: This point is already partly addressed by the motivation given above. Regarding the estimation of the probability of lightning at location A we have to admit that this is also possible by averaging the observed lightning days over all years in the data base and maybe also to include some neighboring locations in order to smooth the estimate. However, things are getting more complex when this has to be applied to several locations simultaneously. This is especially true for regions with complex terrain, where a smoothing is desired not only over space and time but also over the altitude. Estimating climatological values by averaging easily leads to an arbitrary selection of smoothness. Thus our aim is to present a neat method that helps researchers to produce valuable

climatologies from their raw lightning data.

General point 2: In addition, the model is developed using the lightning data itself, and then tries to predict the same lightning data. So the model input and output are not independent of each other. A correct model should use parameters A, B and C to predict D. not A, B and D to predict D. Furthermore, the model should be developed for a specific period, i.e. 1992 to 2000 (for example) and then tested on the year 2001 to see if the model can reproduce the lightning of 2001. In fact, it would be interesting and valuable to compare the model output (2001) with the real data (2001). How well correlated are the lightning estimates by the model for 2001 (based on a model constructed with input data from 1992-2000) with the real lightning from 2001. That is a legitimate test of the model.

Answer: This is absolutely right. We applied this procedure by cross-validation. 6 years of data are available. The parameters of the model are estimated based on 5 years of the data and validated on the remaining year. This is done 6 times with every single year serving as validation period once. All scores presented in the study are based on this procedure which is state-of-the-art in statistics. We will describe this procedure in more detail in the next version of the manuscript.

General point 3: Finally, if the model is a physical model, then it should be applicable to other regions of Austria. How well does this model predict lightning in other regions of Austria (or Europe)? If it is only good for Carinthia, then why bother? Just use the real observed data for risk maps.

Answer: It is not a physical model, but a statistical one. The presented method can still be applied to other regions.

–

Specific comments:

Page 2 Line 2: of Austria vary

Answer: Will be corrected.

Line 24: data are

Answer: Will be corrected.

line 30: period are

Answer: Will be corrected.

Page 3 line 1: What is the detection efficiency of the intra-cloud flashes relative to the CG flashes?

Answer: Because it is impossible to determine the detection efficiency of intra-cloud flashes without a locally installed VHF network (e.g. LMA), there was no attempt made in Schulz (2016) to characterize the detection efficiency of intra-cloud flashes.

line 6: what about the detection efficiency in %?

Answer: Schulz (2016) show that the flash detection efficiency is greater than 96% (100%) if one of the return strokes in a flash had a peak current greater than 2kA (10kA).

line 13: data are

Answer: Will be corrected.

line 24: Is it a correct assumption to assume a random process?

Answer: Yes, it is a general assumption made for statistical modelling.

Page 4: This reminds me of the KISS principle.....(Keep it Simple, Stupid)

Answer: Yes, we tried to keep it simple. However, I agree that the reader might feel overwhelmed. Nevertheless, eq. (3) is important as it introduces the smoothing parameter lambda which is tuned by cross-validation.

Page 5 line 27: The raw data also shows that the main lightning season is from June

until end of August. So what is so great about this model? Why do we need it? To tell us something we already know from the raw data? I don't understand the logic behind the model. What can it tell us that we don't already know.

Answer: Cf. general point 1.

Page 8 line 21: But if we HAVE the climatology, why do we need this tool?

Answer: Cf. general point 1.

line 24: I do not understand why this is a useful too when the raw data give a better estimate of the climatology.

Answer: Cf. general point 1.

line 26: smooth estimates can also be obtained by averaging the raw data temporally and spatially.

Answer: That is true. This procedure would refer to a k-nearest-neighbor estimation. However, one would have to find the optimal width for the smoothing windows in time and space. The analogy between finding the smoothing parameter lambda in a GAM framework and finding the width for a smoothing window is illustrated by Hastie et al. (2009, chapter 6.2).

line 27: Why not simply use the real raw data? I do not understand why a model is needed.

Answer: Cf. general point 1.

Page 12 Figure 2 caption: cells with

Answer: Will be corrected.

Page 13 Figure 3: What are units of y-axis in upper plots?

Answer: No units. The values are on the scale of the additive predictor, i.e. the right hand side of eq. (1).

Page 14 Figure 5: How does this differ from the raw data climatology. Maybe show one next to the other.

Answer: Thanks. That's a good idea. I will consider this and add empirical estimates from the raw data for comparison.

Page 15 Figure 7: What are the numbers in the key of the figure on right. 2.1? 7?.

Answer: The legend shows expected number of flashes accumulated over the lightning season.

–

References:

Hastie, T., Tibshirani, R., and Friedman, J.: The Elements of Statistical Learning (2nd edition), Springer-Verlag, 763 pages, 2009.

Schulz, W., Diendorfer, G., Pedeboy, S., and Poelman, D. R.: The European lightning location system EUCLID – Part 1: Performance analysis and validation, Nat. Hazards Earth Syst. Sci., 16, 595-605, doi:10.5194/nhess-16-595-2016, 2016.

–

---

## Referee Comment (RC2) · Anonymous Referee #2 · 11 Aug 2016

**1    General Comments**

In the study 'Spatio-temporal smoothing of lightning climatologies' Thorsten Simon and colleagues develop methods for estimating the climatological probability of occurrence and the intensity of lightning. For this purpose a GAM was used that is depending on altitude, geographical location and season (day of the year). The developed methods has been applied to the summertime observations between 2010 and 2015 of the lightning location system ALDIS in the Austrian state of Carinthia (Eastern Alps). In this regard, the results for the intensity and the occurrence model are quite similar: the altitude effect shows higher values for higher elevation, the spatial effect provides a minimum in the Northwest and a maximum in the Eastern part of Carinthia, the seasonal effect peaks in mid July. However, the main difference between intensity and

occurrence model can be found in the spatial effect, since a higher spatial variability is provided for the intensity model. This aspect could be explained due to a higher influence of local constructions on the lightning intensity. Furthermore, quantitative applications for the proposed approach are presented.

This paper is basically comprehensible, well structured and written in good English. Moreover, the general idea of the paper is interesting and the given approach is straight forward and certainly viable.

Since I got the impression that a major asset is the modeling for a complex terrain, I would like to know what is the benefit of adding an altitude effect to the statistical model, whereas the lon/lat-part seems to be the most influential effect? Moreover, I am not sure whether spatial function and altitude function are really distinct. Isn't it just sufficient to take the location into account because it implicitly contains the altitude?

Finally, in terms of verification, it is not clear what kind of scores were calculated or used and what their results are.

**2 Specific Comments**

**2.1 Title**

I am afraid that the title 'Spatio-temporal smoothing of lightning climatologies' is misleading, because spatio-temporal smoothing implies some kind of grid-wise and time-wise moving average or filter, while the main idea of your study is to decompose the signal into a seasonal, spatial and also altitude effect by a statistical model. Reading the paper, I would have entitled it something like 'Statistical modeling of lightning climatologies for complex terrains' or 'Spatio-temporal smoothing of lightning climatologies for complex terrains'...

[Figure]

**2.2 Introduction**

Reading your introduction, I got the impression that thunderstorms/lightning tends to occur at regions with moderate or lower altitude (page 2, lines 4-8). But your figures 3 and 4, top-left implying a positive and linear relationship between altitude and occurrence/intensity. Why doesn't the GAM fits a function with maxima for lower/moderate altitudes?

**2.3 Data**

Page 3, line 1: Reading this, with little experience on this scientific field, I would like to know the distinction between lightning, flash and stroke?

Maybe, it would be interesting to show a figure with the spatial climatologies of the number of flashes in Carinthia for the raw data.

**2.4 Methods**

page 4, line 6: As mentioned before, are altitude and horizontal space (lon/lat) really distinct. Thus, eq. (1) probably would have the form: $g(\theta) = \beta_0 + f_1(doy) + f_2(lon, lat, logalt)$

**2.5 Results**

page 5, line 20: How does the 1000 day-wise block-bootstrapping work?
page 6, line 4: Is there any explanation for the maximum in the Gurktal Alps, although this region is quite low elevated?

page 6, line 11: Is there any explanation for the flatter shape of the altitude effect function?

**2.6 Discussion**

In my opinion the part where the authors explain that cross-validation with day-wise blocks is much smoother and subsequently recommend to explore dependence structure of the data first would be more suitable for the method section.

**2.7 Conclusion**

Page 8, line 30-32: As far as I understand, in section 4.2 the higher spatial variability of the intensity model is explained due to local constructions, that trigger the number of flashes without affecting the occurrence. However, in the conclusion part one get the impression that higher spatial variability of the intensity model is distinct from local maxima in the vicinity of radio towers. Thus, I would suggest a sentence like: 'In particular the spatial effect of the intensity model varies more strongly than the corresponding effect of the occurrence model, because local intensity maxima are triggered in vicinity of radio towers. Moreover other new features were exhibited like...'

---

## Author Comment (AC2) · 23 Aug 2016

Answers to Referee 2
* * *
Thank you very much for your informative comments. They will clearly help improving the quality of the manuscript.

General Comments:
* * *
"This paper is basically comprehensible, well structured and written in good English. Moreover, the general idea of the paper is interesting and the given approach is straight forward and certainly viable."

[Figure]

Answer: Thanks for acknowledging this. We agree that the approach is straightforward for someone with experience in GAMs. However, we feel that this is not necessarily the case for all readers of this manuscript at the intersection of lightning science, climatology, and applied statistics. Hence one objective of the manuscript is to bridge some of the gaps and make GAMs more accessible to researchers in the field of lightning science. Both your comments and those of Referee 1 show that we haven't fully accomplished this goal and hence we are grateful for your suggestions for improvements.

"Since I got the impression that a major asset is the modeling for a complex terrain, I would like to know what is the benefit of adding an altitude effect to the statistical model, whereas the lon/lat part seems to be the most influential effect? Moreover, I am not sure whether spatial function and altitude function are really distinct. Isn't it just sufficient to take the location into account because it implicitly contains the altitude?"

Answer: It is true, the altitude is a function of longitude and latitude. In general the presented method would be capable to model the influence of the altitude within the spatial effect implicitly. However, the shape of the altitude in the region of interest is very complex. Thus, a spatial effect with a large degree of freedom would be required in order to account for the complex altitude shape. As we know the shape of the altitude we can pass it to the GAM as an isolated effect. The altitude effect contains only information associated with the altitude while the remaining effects are captured by the lon/lat term. We will mention this aspect in the results section.

"Finally, in terms of verification, it is not clear what kind of scores were calculated or used and what their results are."

Answer: The log-likelihood is applied, also called logarithmic score in the literature on proper scoring rules (see Gneiting 2007). We tried to avoid showing the results of the scores in detail, which would mean showing a longish table with proposed values of the smoothing parameter and associated scores from which the best is selected. Instead we wanted to put more emphasis on the results. We will add a paragraph in

the methods section to discuss the verification score.

A table summarizing the verification scores would look like this:

| $\lambda$ | Q. 2.5% | Median | Q. 97.5% | d.o.f. |
|---|---|---|---|---|
| – | 762455 | 765275 | 768364 | 0.00 |
| 5e+09 | 754052 | 756881 | 759698 | 1.07 |
| 1e+09 | 751785 | 754598 | 757377 | 1.27 |
| 5e+08 | 749880 | 752880 | 755764 | 1.45 |
| 1e+08 | 746750 | 749558 | 752554 | 2.11 |
| **5e+07** | **746356** | **749251** | **752243** | **2.51** |
| 1e+07 | 746754 | 749571 | 752341 | 3.77 |
| 1e+06 | 748266 | 751320 | 754269 | 6.73 |
| 100000 | 753277 | 756236 | 759019 | 11.78 |
| 10000 | 764341 | 767496 | 770352 | 19.38 |
| 0 | 786475 | 789802 | 792789 | 29.00 |

**Table 1.** 6-fold cross-validated negative log-likelihood for different smoothness parameters of the temporal effect. The dash in the $\lambda$-column indicates that no temporal effect was included into the model. Median, $2.5\%$ quantile and $97.5\%$ quantile was generated by bootstrapping $1000$ times.

Specific Comments:

——————

Title "I am afraid that the title 'Spatio-temporal smoothing of lightning climatologies' is misleading, because spatio-temporal smoothing implies some kind of grid-wise and time-wise moving average or filter, while the main idea of your study is to decompose the signal into a seasonal, spatial and also altitude effect by a statistical model. Reading the paper, I would have entitled it something like 'Statistical modeling of lightning climatologies for complex terrains' or 'Spatio-temporal smoothing of lightning climatologies for complex terrains'..."

Answer: Thanks for pointing this out and for suggesting the alternatives. We will change the title to "Spatio-temporal modeling of lightning climatologies for complex terrain".

Introduction "Reading your introduction, I got the impression that thunderstorms/lightning tends to occur at regions with moderate or lower altitude (page 2, lines 4-8). But your figures 3 and 4, top-left implying a positive and linear relationship between altitude and occurrence/intensity. Why doesn't the GAM fits a function with maxima for lower/moderate altitudes?"

Answer: The observed maxima at moderate or lower altitude are not the general case, but special cases associated with local effects. Thus it is not visible in the altitude effect. E.g., the maximum in the Gurktal Alps cannot be explained by its altitude, but the maximum has to be a consequence of local attributes of the terrain in that region. We will add a sentence in the results section where the effect is introduced.

Data "Page 3, line 1: Reading this, with little experience on this scientific field, I would like to know the distinction between lightning, flash and stroke?"

Answer: Lightning is defined as a transient, high-current (typically tens of kiloamperes) electric discharge in the air whose length is measured in kilometers. The lightning discharge in its entirety is usually termed a 'lightning flash' or just a 'flash'. Each flash typically contains several 'strokes' which is the basic element of a lightning discharge. We will add a sentence in the data section to clarify the nomenclature.

"Maybe, it would be interesting to show a figure with the spatial climatologies of the number of flashes in Carinthia for the raw data."

Answer: Such a figure will be added with explainations (cf. Fig. 1 in this response).

Methods "page 4, line 6: As mentioned before, are altitude and horizontal space (lon/lat) really distinct. Thus, eq. (1) probably would have the form: $g(\theta) = \beta_0 + f_1(doy) + f_2(lon, lat, logalt)$"

Answer: As pointed out above one could just use $f_2(lon, lat)$ because logalt is a function of lon/lat but that would necessitate a very complex lon/lat term (using many degrees of freedom). The suggestion $f_2(lon, lat, logalt)$ could be interpreted as a spatially varying logalt effect. This is, in principle, also possible but is also more challenging to estimate. The additive decomposition $f_2(lon, lat) + f_3(logalt)$ is "the usual" trick of using an additive decomposition of the effect which leads to relative parsimonious effects $f_2$ and $f_3$. Of course, there may be even better parametrizations but this seems to work well and is (relatively) easy to interpret for practitioners.

Results "page 5, line 20: How does the 1000 day-wise block-bootstrapping work?"

Answer: With day-wise block-bootstrapping we mean the following: We draw 738 dates of all available days with repetition and pick all the data observed on these days spatially. If we would relax the day-wise structure we would draw 7309152 samples with repetition from all available data points. An explaination will be added in the manuscript.

"page 6, line 4: Is there any explanation for the maximum in the Gurktal Alps, although this region is quite low elevated?"

Answer: We haven't found an explanation yet. However, in a follow-up study we will set the focus on analysis of single events and associated synoptical situations. Hopefully, this study will provide more insights.

"page 6, line 11: Is there any explanation for the flatter shape of the altitude effect function?"

Answer: We haven't found a sound and strong explanation for that shape.

Discussion "In my opinion the part where the authors explain that cross-validation with day-wise blocks is much smoother and subsequently recommend to explore dependence structure of the data first would be more suitable for the method section."

Answer: Yes, we agree that this could be part of the methods section which would probably be the more natural section for readers with experience in flexible regression

modeling (with GAMs). However, we deferred it to the discussion in order to make the methods section more accessible for readers not so familiar with GAMs. Hence we felt it would be easier for that audience if the the cross-validation is explained along the concrete example rather than abstract formulae. To better accomodate readers with experience in GAMs we have now added a forward reference in the methods section with only some short comments.

Conclusion "Page 8, line 30-32: As far as I understand, in section 4.2 the higher spatial variability of the intensity model is explained due to local constructions, that trigger the number of flashes without affecting the occurrence. However, in the conclusion part one get the impression that higher spatial variability of the intensity model is distinct from local maxima in the vicinity of radio towers. Thus, I would suggest a sentence like: 'In particular the spatial effect of the intensity model varies more strongly than the corresponding effect of the occurrence model, because local intensity maxima are triggered in vicinity of radio towers. Moreover other new features were exhibited like...'"

Answer: We adopt this sentence.

References:

————

Gneiting, Tilmann, and Adrian E. Raftery. "Strictly proper scoring rules, prediction, and estimation." Journal of the American Statistical Association 102.477 (2007): 359-378.

[Figure]

**Fig. 1.** Empirical climatological probability of lightning for a day in July in Carinthia on the 1km x 1km scale.

---

## Referee Report (RR1)

**1 General Comments**

After reading the revised manuscript, I got the impression that the authors generally implemented the referee comments satisfactorily. However, some questions and suggestions for improvement still arose.

In the introduction the authors mention, that the main motivation to process raw data by a statistical model is to improve the signal-to-noise ratio. Therefore, I would suggest to show two figures with the spatial distribution of the coefficient of determination $R^2$ for the probability and the intensity of lightning. This may help to visualize the effect of the proposed smoothing and would show how much variance of the observations could be explained by this statistical model.

**2 Specific Comments**

**2.1 Generalized additive models**

page 5, line 1: Is there a relationship between $\lambda$ and the degree of freedom? If there is a relationship, it would be helpful to mention it, because the selection of your $\lambda$ has an impact on your d.o.f., which is (as far as I understand) one of your models main benchmarks.

In terms of d.o.f., it would also be helpful to explain its values, i.e. d.o.f=0 is a linear fit, d.o.f=1 and d.o.f=2 are quadratic and cubic polynomials...

As far as I know, degree of freedom often is defined as the number of independent scores that go into the estimate minus the number of parameters, while you are defining the d.o.f. only as number of parameters. Do I misunderstood sth.?

**2.2 Verification**

page 5, line 28: which parameters were estimated, $\beta_0, \beta_1, ...$ or $\lambda$ or both? At this point I would like to know, how do you estimate $\lambda$? Du you simultaneously estimate $\beta_j$ and $\lambda$ during the training period and try to find an optimum $\beta_j$ and $\lambda$ that minimize your negative maximum likelihood for the validation period? Or do you initially set $\lambda$ to a certain value (e.g. 100000), then estimate $\beta_j$ during the training period and calculate the log-likelihood within your validation period with the estimated $\beta_j$ and the preset $\lambda$?

**2.3 Discussion**

page 9, line 1-3: I got confused by the difference between cross-validation with day-wise blocks and cross-validation without these day-wise blocks. Maybe it would be helpful, if you write that day-wise means cross-validation at every grid point with 6x123 data points/days and without day-wise means cross-validation with every grid point and

every day (in this case 6x123x25 data points). You are explaining this term already in the verification section, but for me it was difficult to transfer from day-wise block bootstrapping to without day-wise cross-validation, since without day-wise could have various meanings.

page 9, line 3: Is there a reason for setting the maximum d.o.f. to 30?

---

## Author Response (AR2)

We are especially grateful for the detailed and informative comments. We think that the quality and readability of the manuscript have been improved once again. Hereby we would like to thank the editor and the referees for their effort, time and thoughts.

We re-structured and partly re-wrote the introduction. From the referees' points we had the impression that the motivation to apply GAMs for such a data set has to be more sound and direct to reach the target readers.

**Response to Editor**

**P1 L21: What other covariates except for altitude may be important? May they be introduced in the model?**

Basically *everything* could be added as a covariate. With respect to climate assessments properties derived from topography and land use attributes are most meaningful. We added some more examples at this point in the introduction. We also tried several different covariates derived from topography. However they did not improve the model. We now mention this in the results section.

**P2 L8ff: Maybe you should not explicitly refer to ALDIS here since the data is only described later in the paper.**

Agree.

**P2 L30: What parameters of a distribution can be specifically modeled?**

Basically all parameters of a distribution can be described by additive models. Often these parameters are associated with the scale and the shape of a distribution.

**P3 L1ff: The first paragraph in this section might be better inserted into the introduction.**

Agree.

**P3 L21: What is the original resolution of the DEM data? How was it acquired?**

Added the original resolution. The data are provided by the federal state of Carinthia. The link to the web page is included in the reference list.

**Figure 1: Please show the position of Carinthia in a location map. The map has no scale (as the other maps).**

The axes are labeled now. If you prefer I can also add a location map.

**Figure 4, 5: Please denote parts as A, B, C and refer in text and caption accordingly.**

Thank you. We changed that.

**Response to Referee #1**

**I have read the revised paper and find that the paper is somewhat improved, but I still have some issues with the methodology and results. 1. The analysis of the lightning data using the GAM is simply a fancy way of smoothing the data spatially and temporally. Such smoothing functions (whether splines or interpolations) are available in most software packages (Matlab, IDL, R, Python, etc.) and hence this is not a new development of the authors. And I still wonder why the authors think their GAM is better than simply smoothing the raw data.**

Thank you for insisting that the advantages of using GAMS instead of averaging per grid cell have not become clear enough yet since the clarity would probably be also found to be insufficient by many readers!

We have completely rewritten the introductory section and also changed several parts of the paper including the conclusion to both motivate and show the advantages of generalized additive models (GAMs). Since lightning is a rare event with 0.5–4 flashes per year and square kilometer (Schulz et al., 2005) in the eastern Alps and available time series are short (on the order of 10 years), the sample size is too small to compute a climatology on such fine scales as $km^{-2}d^{-1}$ by simply averaging the number of flashes in each grid cell. To be able to estimate climatologies on such fine scales, information from the whole data set has to be used instead, which general additive models allow to do. Lightning e.g., might depend on altitude so that combining the information from all cells at a particular altitude band will increase the sample size and thus lead to a more robust estimate. GAMs also permit to introduce expert knowledge to refine the climatology, e.g., altitude, aspect of the slope, geographical location, soil moisture. Importantly, GAMS allow to also test which of the proposed effects is significant and thus an actual effect. An even further advantage is the ability to obtain not only expected values (means) but also the full probability distribution, which is highly skewed in the case of lightning. We derived the relative frequencies for the sample locations for a single day and present these as additional application.

**I agree that the observed data is noisy, but maybe for a reason. Maybe the noisy topography in the region results in peaks of lightning above mountain peaks, less in valleys, and when the model smooths the data these maxima disappear. Smoother data is not always better or closer to reality. Maybe the noisier real data is better for determining risks.**

The difference between lightning near peaks and over valleys that you mention becomes obvious using GAMs while it is obscured by a the grid-cell-averaging method as can be seen by comparing Figs. 6 and 3.

**2. The authors use 3 parameters only to describe the lightning climatologies (altitude, day of year, and longitude/latitude). However, in reality there may be many other parameters that determine the lightning distribution. For example, vegetation cover, slope of topography, soil moisture, etc. Hence, the model can only be as good as the input parameters.**

Yes, that is one advantage of GAMs. Indeed we tried some other covariates (input parameters), e.g., surface roughness, aspect and slope of topography. However, they did not come with an effect. We now mention this in the result section. However, if there would a covariate which would have an effect and could be written as a function of long/lat, its effect would implicitly included in the spatial effect. Adding this covariate would then

lead to a smoother spatial effect. This property of the GAM is already explained in the text (results section, occurrence model).

**At the bottom of P5, the authors state that the model is generated with 5 years of data, and then tested against the 6th year of data. However, I do not see these comparisons. Where is the predicted distribution for year XXXX next to the observed (smoothed) distribution for year XXXX. Such comparisons are necessary to show that the three parameters used are sufficient for building the climatology.**

Cross-validation is a standard method for testing a statistical model on independent data. The basic idea of the cross-validation is to train the model on 5 years of data and validate the model on the remaining year. The validation is expressed in a score, i.e., the log-likelihood. This procedure is repeated 6 times in such way that every year serves as validation period once. In the end the 6 scores are summed up to express the out-of-sample performance of the model. The cross-validation is applied in order to determine the best values for the smoothing parameters $\lambda_j$. Thus the visual comparison between the model trained on 5 years and the observations of the 6th year is not subject of the cross-validation, instead quantitative methods have been applied.

**Minor comments:**

**P1 line 12: Simply smooting the observed data would also produce a climatology that varies smoothly over space and time. This is not unique to your method.**

Right. However, GAMs provide a statistical model which can be analyzed quantify and GAMs can be easily extended with other covariates. We added that the climatology resulting from our GAM varies also smoothly over the altitude. For instance, it is not clear to me how to filter the altitude effect with simply smoothing.

**P2 line 21-23: Why is using the raw data a problem for quantitative assessments? The same method can be followed to assess the risk using the raw data, however noisy is may be. You can use simple spline or interpolation to smooth the observed data and get similar results.**

The raw data are too numerous for any kind of quantitative assessment, but one has to apply some kind of descriptive statistical analysis to it in order to receive the information sought after. Within our GAM we are using splines. We think that it is a very good tool to learn from the data, e.g., we did not only see a smooth pattern in space and time, but a also receive a quantification of the altitude effect. Potentially other covariates could be employed. Moreover, the selection of the complexity of the model by cross-validation is an objective way. We left this part of the text the way it was, but tried to motivate the usage of GAMs a bit more in the two consecutive paragraphs and added a paragraph in the section 4.3 Applications.

**P3 line 19: Why was only Carinthia used in this study, and not the whole of Austria. I guess the data is available, so why not use it?**

Carinthia is the area with the strongest lightning activity within Austria. Thus most interesting to investigate. Preprocessing the data to the $km^{-2}d^{-1}$ scale leads to roughly 7 million data points. Fitting a model on my local machine takes less than 10 min. In order to provide confidence interval we run the bootstrapping (resampling the data and fitting the model 1000 times) parallel on the HPC infrastructure LEO of the University of Innsbruck. Therefore the study is overall computationally demanding. We think the study in the present way highlights all

important aspects and potential of the method. Taking the computational effort and estimating a climatology for the whole of Austria at the same resolution will not add too much value.

**P3 line 29: Is this the number of flashes over the 4 month period? Please clarify.**

Yes. We repeated that the data is from observations during summer (May to August) of 6 years.

**P6 line 11: Fig.4 is still not clear regarding the y-axis units. Are these the weighting functions? Are they dimensionless? What is the physical meaning of 0.5? What is the physical meaning of a negative value? Please explain. If it is not clear for me, I guess others will have difficulty as well.**

Thank you for pointing out that this might be unclear. We added a paragraph in the results section to illustrate the interpretation of the effects.

**P6 line 22: We don't need the model (or the data) to tell us that lightning is mainly in the summer in Austria. I think this is well known.**

Right. This finding is not surprising. However, this finding is not the central statement of the manuscript, but only mentioned in one sentence for the sake of completeness.

**P6 line 28: If we already know all of this from previous studies, do we need another paper to make this statement? It is difficult for me to figure out what is new in this analysis.**

The previous study analyzed lightning detection data from a different period. We think it is important and interesting to see whether the results match or not.

**P9 line 21: The prediction tool will not fall below the climatology only if all relevant parameters are included in your model. As mentioned above you only used 3 parameters. And we have not seen any comparison between lightning predicted using your model for say, 2015, compared with the observed climatology for 2015. Please present such a comparison, including the correlation coefficient between the predicted and observed distributions.**

We assume that something got misunderstood here and we try to clarify it. By *prediction tool* we did not mean the model presented, but a potential extension. The presented model characterizes the lightning climatology. If the climatology model,

$$g(\theta) = \beta_0 + f_1(\texttt{logalt}) + f_2(\texttt{doy}) + f_3(\texttt{lon}, \texttt{lat}), \tag{1}$$

is nested within a weather prediction model, e.g.,

$$g(\theta) = \beta_0 + f_1(\texttt{logalt}) + f_2(\texttt{doy}) + f_3(\texttt{lon}, \texttt{lat}) + f_4(\texttt{cape}), \tag{2}$$

where cape could be the convective available potential energy taken from a numerical weather prediction system, e.g., ECMWF HRES. In such a case the performance of the weather prediction model would not fall below the performance of the climatology model by construction.

**Response to Referee #2**

**1 General Comments**

**After reading the revised manuscript, I got the impression that the authors generally implemented the referee comments satisfactorily. However, some questions and suggestions for improvement still arose.**

Motivated by your comment and also the other reviewer's comments we have completely rewritten the introductory section that motivates the need for a method that can harness the information in the complete data instead of just taking the average locally in each grid cell. For such fine scales as $km^{-2}d^{-1}$, the sample size for estimating a climatology by taking averages is to small given typical rates of a few flashes per $km^2$ and year and typical lightning data set lengths of about 10 years. Further advantages are the ability to include expert knowledge for the refinement of the climatologies and the ability to test which parts of the expert knowledge contribute significantly to improving the climatology. Parts of the paper including the conclusion have also been written in order to more clearly show the advantages of using GAMs.

**In the introduction the authors mention, that the main motivation to process raw data by a statistical model is to improve the signal-to-noise ratio. Therefore, I would suggest to show two figures with the spatial distribution of the coefficient of determination $R^2$ for the probability and the intensity of lightning. This may help to visualize the effect of the proposed smoothing and would show how much variance of the observations could be explained by this statistical model.**

A map would not be appropriate at his point, rather one can analyze these values for the whole model. In terms of explained deviance the occurrence model and the intensity model reach 6.8% and 3.5%, respectively—in terms of adjusted $R^2$ 4.3% and 0.8%. As we stated in the introduction the lightning data is very noise by nature. In this light it was expected that the model explains only a small part of the variability. The clear benefit is that the signals/effects for time, space and altitude are well separated form the noise.

**2 Specific Comments**

**2.1 Generalized additive models**

**page 5, line 1: Is there a relationship between $\lambda$ and the degree of freedom? If there is a relationship, it would be helpful to mention it, because the selection of your $\lambda$ has an impact on your d.o.f., which is (as far as I understand) one of your models main benchmarks. In terms of d.o.f., it would also be helpful to explain its values, i.e. d.o.f=0 is a linear fit, d.o.f=1 and d.o.f=2 are quadratic and cubic polynomials . . .**

Right. There is a relationship between $\lambda$ and the d.o.f. However, this relationship can not be expressed by a formula. The table in first rejoinder showed one example for this relationship, but we think that these numbers would not add too much value to the paper as this is quite technical. However we added some more explanations to the methods section.

**As far as I know, degree of freedom often is defined as the number of independent scores that go into the estimate minus the number of parameters, while you are defining the d.o.f. only as number of parameters. Do I misunderstood sth.?**

Right. In a classical linear model—without penalization—the d.o.f. is equal to the number of coefficients to be estimated. Or speaking technically: The trace of the hat matrix H is equal to its rank, where H is defined by

$$\hat{y} = Hy = X(X^\top X)^{-1}X^\top y, \tag{3}$$

with X, y and $\hat{y}$ denoting the design matrix, the response vector and its estimates, respectively. Here the trace of H is equal to the degrees of freedom of the linear model. With penalization the estimates are

$$\tilde{y} = \tilde{H}y = X(X^\top X + S)^{-1}X^\top y, \tag{4}$$

where $\tilde{y}$ are the estimates of the penalized regression and S is the penalty matrix (cf. Eq. 3 in the manuscript). Again the degrees of freedom is defined as the trace of the matrix $\tilde{H}$, though it is no longer equal to the number of coefficients. We think that all this is far too technical for the paper. However, the interested reader is referred to the textbook by Wood.

**2.2 Verification**

**page 5, line 28: which parameters were estimated, $\beta_0, \beta_1, \ldots$ or $\lambda$ or both? At this point I would like to know, how do you estimate $\lambda$? Du you simultaneously estimate $\beta_j$ and $\lambda$ during the training period and try to find an optimum $\beta_j$ and $\lambda$ that minimize your negative maximum likelihood for the validation period? Or do you initially set $\lambda$ to a certain value (e.g. 100000), then estimate $\beta_j$ during the training period and calculate the log-likelihood within your validation period with the estimated $\beta_j$ and the preset $\lambda$?**

For a single $\lambda$ a set of $\beta_0, \beta_1, \ldots$ can be estimated. However, as explain in the newly added paragraph in the method section, the value of $\lambda$ determines the smoothness of the associated effect. Cross-validation is applied to select the value of $\lambda$.

**2.3 Discussion**

**page 9, line 1-3: I got confused by the difference between cross-validation with day-wise blocks and cross-validation without these day-wise blocks. Maybe it would be helpful, if you write that day-wise means cross-validation at every grid point with $6 \times 123$ data points/days and without day-wise means cross-validation with every grid point and every day (in this case 6x123x25 data points). You are explaining this term already in the verification section, but for me it was difficult to transfer from day-wise block bootstrapping to without day-wise cross-validation, since without day-wise could have various meanings.**

Thank you for pointing at the confusion. We extended the explanation at bit at this point in order to clarify this issue.

**page 9, line 3: Is there a reason for setting the maximum d.o.f. to 30?**

The choice of the maximum is kind of arbitrary. However, it is important that one allows the effect to be sufficiently flexible. For an annual cycle, like in this case, one would expect the d.o.f. to fall below 10. Thus one could also set the maximum d.o.f. to 20, 40 or 100.

[revised manuscript text omitted]